# Deeply Learning the Messages in Message Passing Inference

**Guosheng Lin, Chunhua Shen, Ian Reid, Anton van den Hengel**
The University of Adelaide, Australia; and Australian Centre for Robotic Vision
E-mail: {guosheng.lin,chunhua.shen,ian.reid,anton.vandenhengel}@adelaide.edu.au

## Abstract

Deep structured output learning shows great promise in tasks like semantic image segmentation. We proffer a new, efficient deep structured model learning scheme, in which we show how deep Convolutional Neural Networks (CNNs) can be used to directly estimate the messages in message passing inference for structured prediction with Conditional Random Fields (CRFs). With such CNN message estimators, we obviate the need to learn or evaluate potential functions for message calculation. This confers significant efficiency for learning, since otherwise when performing structured learning for a CRF with CNN potentials it is necessary to undertake expensive inference for every stochastic gradient iteration. The network output dimension of message estimators is the same as the number of classes, rather than exponentially growing in the order of the potentials. Hence it is more scalable for cases that involve a large number of classes. We apply our method to semantic image segmentation and achieve impressive performance, which demonstrates the effectiveness and usefulness of our CNN message learning method.

## 1 Introduction

Learning deep structured models has attracted considerable research attention recently. One popular approach to deep structured model is formulating conditional random fields (CRFs) using deep Convolutional Neural Networks (CNNs) for the potential functions. This combines the power of CNNs for feature representation learning and of the ability for CRFs to model complex relations. The typical approach for the joint learning of CRFs and CNNs [1, 2, 3, 4, 5], is to learn the CNN potential functions by optimizing the CRF objective, e.g., maximizing the log-likelihood. The CNN and CRF joint learning has shown impressive performance for semantic image segmentation.

For the joint learning of CNNs and CRFs, stochastic gradient descent (SGD) is typically applied for optimizing the conditional likelihood. This approach requires the marginal inference for calculating the gradient. For loopy graphs, marginal inference is generally expensive even when using approximate solutions. Given that learning the CNN potential functions typically requires a large number of gradient iterations, repeated marginal inference would make the training intractably slow. Applying an approximate training objective is a solution to avoid repeat inference; pseudo-likelihood learning [6] and piecewise learning [7, 3] are examples of this kind of approach. In this work, we advocate a new direction for efficient deep structured model learning.

In conventional CRF approaches, the final prediction is the result of inference based on the learned potentials. However, our ultimate goal is the final prediction (not the potentials themselves), so we propose to directly optimize the inference procedure for the final prediction. Our focus here is on the extensively studied message passing based inference algorithms. As discussed in [8], we can directly learn message estimators to output the required messages in the inference procedure, rather

than learning the potential functions as in conventional CRF learning approaches. With the learned message estimators, we then obtain the final prediction by performing message passing inference.

Our main contributions are as follows:

1) We explore a new direction for efficient deep structured learning. We propose to *directly learn the messages in message passing inference as training deep CNNs in an end-to-end learning fashion.* Message learning does not require any inference step for the gradient calculation, which allows efficient training. Furthermore, when cast as a tradiational classification task, the network output dimension for message estimation is the same as the number of classes ($K$), while the network output for general CNN potential functions in CRFs is $K^a$, which is exponential in the order ($a$) of the potentials (for example, $a = 2$ for pairwise potentials, $a = 3$ for triple-cliques, etc). Hence CNN based message learning has significantly fewer network parameters and thus is more scalable, especially in cases which involve a large number of classes.

2) The number of iterations in message passing inference can be explicitly taken into consideration in the message learning procedure. In this paper, we are particularly interested in learning messages that are able to offer high-quality CRF prediction results with only one message passing iteration, making the message passing inference very fast.

3) We apply our method to semantic image segmentation on the PASCAL VOC 2012 dataset and achieve impressive performance.

**Related work** Combining the strengths of CNNs and CRFs for segmentation has been explored in several recent methods. Some methods resort to a simple combination of CNN classifiers and CRFs without joint learning. DeepLab-CRF in [9] first train fully CNN for pixel classification and applies a dense CRF [10] method as a post-processing step. Later the method in [2] extends DeepLab by jointly learning the dense CRFs and CNNs. RNN-CRF in [1] also performs joint learning of CNNs and the dense CRFs. They implement the mean-field inference as Recurrent Neural Networks which facilitates the end-to-end learning. These methods usually use CNNs for modelling the unary potentials only. The work in [3] trains CNNs to model both the unary and pairwise potentials in order to capture contextual information. Jointly learning CNNs and CRFs has also been explored for other applications like depth estimation [4, 11]. The work in [5] explores joint training of Markov random fields and deep networks for predicting words from noisy images and image classification.

All these above-mentioned methods that combine CNNs and CRFs are based upon conventional CRF approaches. They aim to jointly learn or incorporate pre-trained CNN potential functions, and then perform inference/prediction using the potentials. In contrast, our method here directly learns CNN message estimators for the message passing inference, rather than learning the potentials.

The inference machine proposed in [8] is relevant to our work in that it has discussed the idea of directly learning message estimators instead of learning potential functions for structured prediction. They train traditional logistic regressors with hand-crafted features as message estimators. Motivated by the tremendous success of CNNs, we propose to train deep CNNs based message estimators in an end-to-end learning style without using hand-crafted features. Unlike the approach in [8] which aims to learn *variable-to-factor* message estimators, our proposed method aims to learn the *factor-to-variable* message estimators. Thus we are able to naturally formulate the variable marginals – which is the ultimate goal for CRF inference – as the training objective (see Sec. 3.3). The approach in [12] jointly learns CNNs and CRFs for pose estimation, in which they learn the marginal likelihood of body parts but ignore the partition function in the likelihood. Message learning is not discussed in that work, and the exact relationship between this pose estimation approach and message learning remains unclear.

## 2 Learning CRF with CNN potentials

Before describing our message learning method, we review the CRF-CNN joint learning approach and discuss limitations. An input image is denoted by $x \in \mathcal{X}$ and the corresponding labeling mask is denoted by $y \in \mathcal{Y}$. The energy function is denoted by $E(y, x)$, which measures the score of the prediction $y$ given the input image $x$. We consider the following form of conditional likelihood:

$$P(y|x) = \frac{1}{Z(x)} \exp\left[-E(y, x)\right] = \frac{\exp\left[-E(y, x)\right]}{\sum_{y'} \exp\left[-E(y', x)\right]}. \tag{1}$$

Here $Z$ is the partition function. The CRF model is decomposed by a factor graph over a set of factors $\mathcal{F}$. Generally, the energy function is written as a sum of potential functions (factor functions):

$$E(\boldsymbol{y}, \boldsymbol{x}) = \sum_{F \in \mathcal{F}} E_F(\boldsymbol{y}_F, \boldsymbol{x}_F). \qquad (2)$$

Here $F$ indexes one factor in the factor graph; $\boldsymbol{y}_F$ denotes the variable nodes which are connected to the factor $F$; $E_F$ is the (log-) potential function (factor function). The potential function can be a unary, pairwise, or high-order potential function. The recent method in [3] describes examples of constructing general CNN based unary and pairwise potentials.

Take semantic image segmentation as an example. To predict the pixel labels of a test image, we can find the mode of the joint label distribution by solving the maximum a posteriori (MAP) inference problem: $\boldsymbol{y}^\star = \mathrm{argmax}_{\boldsymbol{y}} P(\boldsymbol{y}|\boldsymbol{x})$. We can also obtain the final prediction by calculating the label marginal distribution of each variable, which requires to solve a marginal inference problem:

$$\forall p \in \mathcal{N}: \quad P(y_p|\boldsymbol{x}) = \sum_{\boldsymbol{y} \setminus y_p} P(\boldsymbol{y}|\boldsymbol{x}). \qquad (3)$$

Here $\boldsymbol{y} \setminus y_p$ indicates the output variables $\boldsymbol{y}$ excluding $y_p$. For a general CRF graph with cycles, the above inference problems is known to be NP-hard, thus approximate inference algorithms are applied. Message passing is a type of widely applied algorithms for approximate inference: loopy belief propagation (BP) [13], tree-reweighted message passing [14] and mean-field approximation [13] are examples of the message passing methods.

CRF-CNN joint learning aims to learn CNN potential functions by optimizing the CRF objective, typically, the negative conditional log-likelihood, which is:

$$-\log P(\boldsymbol{y}|\boldsymbol{x}; \boldsymbol{\theta}) = E(\boldsymbol{y}, \boldsymbol{x}; \boldsymbol{\theta}) + \log Z(\boldsymbol{x}; \boldsymbol{\theta}). \qquad (4)$$

The energy function $E(\boldsymbol{y}, \boldsymbol{x})$ is constructed by CNNs, for which all the network parameters are denoted by $\boldsymbol{\theta}$. Adding regularization, minimizing negative log-likelihood for CRF learning is:

$$\min_{\boldsymbol{\theta}} \frac{\lambda}{2} \|\boldsymbol{\theta}\|_2^2 + \sum_{i=1}^{N} [E(\boldsymbol{y}^{(i)}, \boldsymbol{x}^{(i)}; \boldsymbol{\theta}) + \log Z(\boldsymbol{x}^{(i)}; \boldsymbol{\theta})]. \qquad (5)$$

Here $\boldsymbol{x}^{(i)}, \boldsymbol{y}^{(i)}$ denote the $i$-th training image and its segmentation mask; $N$ is the number of training images; $\lambda$ is the weight decay parameter. We can apply stochastic gradient descent (SGD) to optimize the above problem for learning $\boldsymbol{\theta}$. The energy function $E(\boldsymbol{y}, \boldsymbol{x}; \boldsymbol{\theta})$ is constructed from CNNs, and its gradient $\nabla_{\boldsymbol{\theta}} E(\boldsymbol{y}, \boldsymbol{x}; \boldsymbol{\theta})$ can be easily computed by applying the chain rule as in conventional CNNs. However, the partition function $Z$ brings difficulties for optimization. Its gradient is:

$$\nabla_{\boldsymbol{\theta}} \log Z(\boldsymbol{x}; \boldsymbol{\theta}) = \sum_{\boldsymbol{y}} \frac{\exp\left[-E(\boldsymbol{y}, \boldsymbol{x}; \boldsymbol{\theta})\right]}{\sum_{\boldsymbol{y}'} \exp\left[-E(\boldsymbol{y}', \boldsymbol{x}; \boldsymbol{\theta})\right]} \nabla_{\boldsymbol{\theta}} [-E(\boldsymbol{y}, \boldsymbol{x}; \boldsymbol{\theta})]$$
$$= -\mathbb{E}_{\boldsymbol{y} \sim P(\boldsymbol{y}|\boldsymbol{x}; \boldsymbol{\theta})} \nabla_{\boldsymbol{\theta}} E(\boldsymbol{y}, \boldsymbol{x}; \boldsymbol{\theta}). \qquad (6)$$

Direct calculation of the above gradient is computationally infeasible for general CRF graphs. Usually it is necessary to perform approximate marginal inference to calculate the gradients at each SGD iteration [13]. However, repeated marginal inference can be extremely expensive, as discussed in [3]. CNN training usually requires a huge number of SGD iterations (hundreds of thousands, or even millions), hence this inference based learning approach is in general not scalable or even infeasible.

## 3 Learning CNN message estimators

In conventional CRF approaches, the potential functions are first learned, and then inference is performed based on the learned potential functions to generate the final prediction. In contrast, our approach directly optimizes the inference procedure for final prediction. We propose to learn CNN estimators to directly output the required intermediate values in an inference algorithm.

Here we focus on the message passing based inference algorithm which has been extensively studied and widely applied. In the CRF prediction procedure, the "message" vectors are recursively calculated based on the learned potentials. We propose to construct and learn CNNs to directly estimate these messages in the message passing procedure, rather than learning the potential functions. In particular, we directly learn factor-to-variable message estimators. Our message learning framework

is general and can accommodate all message passing based algorithms such as loopy belief propagation (BP) [13], mean-field approximation [13] and their variants. Here we discuss using loopy BP for calculating variable marginals. As shown by Yedidia et al. [15], loopy BP has a close relation with Bethe free energy approximation.

Typically, the message is a $K$-dimensional vector ($K$ is the number of classes) which encodes the information of the label distribution. For each variable-factor connection, we need to recursively compute the variable-to-factor message: $\boldsymbol{\beta}_{p \to F} \in \mathbb{R}^K$, and the factor-to-variable message: $\boldsymbol{\beta}_{F \to p} \in \mathbb{R}^K$. The unnormalized variable-to-factor message is computed as:

$$\bar{\boldsymbol{\beta}}_{p \to F}(y_p) = \sum_{F' \in \mathcal{F}_p \backslash F} \boldsymbol{\beta}_{F' \to p}(y_p). \tag{7}$$

Here $\mathcal{F}_p$ is a set of factors connected to the variable $p$; $\mathcal{F}_p \backslash F$ is the set of factors $\mathcal{F}_p$ excluding the factor $F$. For loopy graphs, the variable-to-factor message is normalized at each iteration:

$$\boldsymbol{\beta}_{p \to F}(y_p) = \log \frac{\exp \bar{\boldsymbol{\beta}}_{p \to F}(y_p)}{\sum_{y'_p} \exp \bar{\boldsymbol{\beta}}_{p \to F}(y'_p)}. \tag{8}$$

The factor-to-variable message is computed as:

$$\boldsymbol{\beta}_{F \to p}(y_p) = \log \sum_{\boldsymbol{y}'_F \backslash y'_p, y'_p = y_p} \exp \left[ - E_F(\boldsymbol{y}'_F) + \sum_{q \in \mathcal{N}_F \backslash p} \boldsymbol{\beta}_{q \to F}(y'_q) \right]. \tag{9}$$

Here $\mathcal{N}_F$ is a set of variables connected to the factor $F$; $\mathcal{N}_F \backslash p$ is the set of variables $\mathcal{N}_F$ excluding the variable $p$. Once we get all the factor-to-variable messages of one variable node, we are able to calculate the marginal distribution (beliefs) of that variable:

$$P(y_p | \boldsymbol{x}) = \sum_{\boldsymbol{y} \backslash y_p} P(\boldsymbol{y} | \boldsymbol{x}) = \frac{1}{Z_p} \exp \left[ \sum_{F \in \mathcal{F}_p} \boldsymbol{\beta}_{F \to p}(y_p) \right], \tag{10}$$

in which $Z_p$ is a normalizer: $Z_p = \sum_{y_p} \exp \left[ \sum_{F \in \mathcal{F}_p} \boldsymbol{\beta}_{F \to p}(y_p) \right]$.

### 3.1 CNN message estimators

The calculation of factor-to-variable message $\boldsymbol{\beta}_{F \to p}$ depends on the variable-to-factor messages $\boldsymbol{\beta}_{p \to F}$. Substituting the definition of $\boldsymbol{\beta}_{p \to F}$ in (8), $\boldsymbol{\beta}_{F \to p}$ can be re-written as:

$$\boldsymbol{\beta}_{F \to p}(y_p) = \log \sum_{\boldsymbol{y}'_F \backslash y'_p, y'_p = y_p} \exp \left\{ - E_F(\boldsymbol{y}'_F) + \sum_{q \in \mathcal{N}_F \backslash p} \left[ \log \frac{\exp \bar{\boldsymbol{\beta}}_{q \to F}(y'_q)}{\sum_{y''_q} \exp \bar{\boldsymbol{\beta}}_{q \to F}(y''_q)} \right] \right\}$$

$$= \log \sum_{\boldsymbol{y}'_F \backslash y'_q, y'_p = y_p} \exp \left\{ - E_F(\boldsymbol{y}'_F) + \sum_{q \in \mathcal{N}_F \backslash p} \left[ \log \frac{\exp \sum_{F' \in \mathcal{F}_q \backslash F} \boldsymbol{\beta}_{F' \to q}(y'_q)}{\sum_{y''_q} \exp \sum_{F' \in \mathcal{F}_q \backslash F} \boldsymbol{\beta}_{F' \to q}(y''_q)} \right] \right\} \tag{11}$$

Here $q$ denotes the variable node which is connected to the node $p$ by the factor $F$ in the factor graph. We refer to the variable node $q$ as a neighboring node of $q$. $\mathcal{N}_F \backslash p$ is a set of variables connected to the factor $F$ excluding the node $p$. Clearly, for a pairwise factor which only connects to two variables, the set $\mathcal{N}_F \backslash p$ only contains one variable node. The above equations show that the factor-to-variable message $\bar{\boldsymbol{\beta}}_{F \to p}$ depends on the potential $E_F$ and $\boldsymbol{\beta}_{F' \to q}$. Here $\boldsymbol{\beta}_{F' \to q}$ is the factor-to-variable message which is calculated from a neighboring node $q$ and a factor $F' \neq F$.

Conventional CRF learning approaches learn the potential function then follow the above equations to compute the messages for calculating marginals. As discussed in [8], given that the goal is to estimate the marginals, it is not necessary to exactly follow the above equations, which involve learning potential functions, to calculate messages. We can directly learn message estimators, rather than indirectly learning the potential functions as in conventional methods.

Consider the calculation in (11). The message $\boldsymbol{\beta}_{F \to p}$ depends on the observation $\boldsymbol{x}_{pF}$ and the messages $\boldsymbol{\beta}_{F' \to q}$. Here $\boldsymbol{x}_{pF}$ denotes the observations that correspond to the node $p$ and the factor $F$. We are able to formulate a factor-to-variable message estimator which takes $\boldsymbol{x}_{pF}$ and $\boldsymbol{\beta}_{F' \to q}$ as

inputs and outputs the message vector, and we directly learn such estimators. Since one message $\boldsymbol{\beta}_{F \to p}$ depends on a number of previous messages $\boldsymbol{\beta}_{F' \to q}$, we can formulate a sequence of message estimators to model the dependence. Thus the output from a previous message estimator will be the input of the following message estimator.

There are two message passing strategies for loopy BP: synchronous and asynchronous passing. We here focus on the synchronous message passing, for which all messages are computed before passing them to the neighbors. The synchronous passing strategy results in much simpler message dependences than the asynchronous strategy, which simplifies the training procedure. We define one inference iteration as one pass of the graph with the synchronous passing strategy.

We propose to learn CNN based factor-to-variable message estimator. The message estimator models the interaction between neighboring variable nodes. We denote by $M$ a message estimator. The factor-to-variable message is calculated as:

$$\boldsymbol{\beta}_{F \to p}(y_p) = M_F(\boldsymbol{x}_{pF}, \mathbf{d}_{pF}, y_p). \tag{12}$$

We refer to $\mathbf{d}_{pF}$ as the dependent message feature vector which encodes all dependent messages from the neighboring nodes that are connected to the node $p$ by $F$. Note that the dependent messages are the output of message estimators at the previous inference iteration. In the case of running only one message passing iteration, there are no dependent messages for $M_F$, and thus we do not need to incorporate $\mathbf{d}_{pF}$. To have a general exposition, we here describe the case of running arbitrarily many inference iterations.

We can choose any effective strategy to generate the feature vector $\mathbf{d}_{pF}$ from the dependent messages. Here we discuss a simple example. According to (11), we define the feature vector $\mathbf{d}_{pF}$ as a $K$-dimensional vector which aggregates all dependent messages. In this case, $\mathbf{d}_{pF}$ is computed as:

$$\mathbf{d}_{pF}(y) = \sum_{q \in \mathcal{N}_F \backslash p} \left[ \log \frac{\exp \sum_{F' \in \mathcal{F}_q \backslash F} M_{F'}(\boldsymbol{x}_{qF'}, \mathbf{d}_{qF'}, y)}{\sum_{y'} \exp \sum_{F' \in \mathcal{F}_q \backslash F} M_{F'}(\boldsymbol{x}_{qF'}, \mathbf{d}_{qF'}, y')} \right]. \tag{13}$$

With the definition of $\mathbf{d}_{pF}$ in (13) and $\boldsymbol{\beta}_{F \to p}$ in (12), it clearly shows that the message estimation requires evaluating a sequence of message estimators. Another example is to concatenate all dependent messages to construct the feature vector $\mathbf{d}_{pF}$.

There are different strategies to formulate the message estimators in different iterations. One strategy is using the same message estimator across all inference iterations. In this case the message estimator becomes a recursive function, and thus the CNN based estimator becomes a recurrent neural network (RNN). Another strategy is to formulate different estimator for each inference iteration.

### 3.2 Details for message estimator networks

We formulate the estimator $M_F$ as a CNN, thus the estimation is the network outputs:

$$\boldsymbol{\beta}_{F \to p}(y_p) = M_F(\boldsymbol{x}_{pF}, \mathbf{d}_{pF}, y_p; \boldsymbol{\theta}_F) = \sum_{k=1}^{K} \delta(k = y_p) z_{pF,k}(\boldsymbol{x}, \mathbf{d}_{pF}; \boldsymbol{\theta}_F). \tag{14}$$

Here $\boldsymbol{\theta}_F$ denotes the network parameter which we need to learn. $\delta(\cdot)$ is the indicator function, which equals 1 if the input is true and 0 otherwise. We denote by $\boldsymbol{z}_{pF} \in \mathbb{R}^K$ as the $K$-dimensional output vector ($K$ is the number of classes) of the message estimator network for the node $p$ and the factor $F$; $z_{pF,k}$ is the $k$-th value in the network output $\boldsymbol{z}_{pF}$ corresponding to the $k$-th class.

We can consider any possible strategies for implementing $\boldsymbol{z}_{pF}$ with CNNs. For example, we here describe a strategy which is analogous to the network design in [3]. We denote by $C^{(1)}$ as a fully convolutional network (FCNN) [16] for convolutional feature generation, and $C^{(2)}$ as a traditional fully connected network for message estimation.

Given an input image $\boldsymbol{x}$, the network output $C^{(1)}(\boldsymbol{x}) \in \mathbb{R}^{N_1 \times N_2 \times r}$ is a convolutional feature map, in which $N_1 \times N_2 = N$ is the feature map size and $r$ is the dimension of one feature vector. Each spatial position (each feature vector) in the feature map $C^{(1)}(\boldsymbol{x})$ corresponds to one variable node in the CRF graph. We denote by $C^{(1)}(\boldsymbol{x}, p) \in \mathbb{R}^r$, the feature vector corresponding to the variable node $p$. Likewise, $C^{(1)}(\boldsymbol{x}, \mathcal{N}_F \backslash p) \in \mathbb{R}^r$ is the averaged vector of the feature vectors that correspond to the set of nodes $\mathcal{N}_F \backslash p$. Recall that $\mathcal{N}_F \backslash p$ is a set of nodes connected by the factor $F$ excluding the node $p$. For pairwise factors, $\mathcal{N}_F \backslash p$ contains only one node.

We construct the feature vector $\boldsymbol{z}_{pF}^{C^{(1)}} \in \mathbb{R}^{2r}$ for the node-factor pair $(p, F)$ by concatenating $C^{(1)}(\boldsymbol{x}, p)$ and $C^{(1)}(\boldsymbol{x}, \mathcal{N}_F \backslash p)$. Finally, we concatenate the node-factor feature vector $\boldsymbol{z}_{pF}^{C^{(1)}}$ and the dependent message feature vector $\mathbf{d}_{pF}$ as the input for the second network $C^{(2)}$. Thus the input dimension for $C^{(2)}$ is $(2r + K)$. For running only one inference iteration, the input for $C^{(2)}$ is $\boldsymbol{z}_{pF}^{C^{(1)}}$ alone. The final output from the second network $C^{(2)}$ is the $K$-dimensional message vector $\boldsymbol{z}_{pF}$. To sum up, we generate the final message vector $\boldsymbol{z}_{pF}$ as:

$$\boldsymbol{z}_{pF} = C^{(2)}\{\ [\ C^{(1)}(\boldsymbol{x}, p)^\top;\ C^{(1)}(\boldsymbol{x}, \mathcal{N}_F \backslash p)^\top;\ \mathbf{d}_{pF}^\top\ ]^\top\ \}. \tag{15}$$

For a general CNN based potential function in conventional CRFs, the potential network is usually required to have a large number of output units (exponential in the order of the potentials). For example, it requires $K^2$ ($K$ is the number of classes) outputs for the pairwise potentials [3]. A large number of output units would significantly increase the number of network parameters. It leads to expensive computations and tends to over-fit the training data. In contrast, for learning our CNN message estimator, we only need to formulate $K$ output units for the network. Clearly it is more scalable in the cases of a large number of classes.

### 3.3 Training CNN message estimators

Our goal is to estimate the variable marginals in (3), which can be re-written with the estimators:

$$P(y_p|\boldsymbol{x}) = \sum_{\boldsymbol{y} \backslash y_p} P(\boldsymbol{y}|\boldsymbol{x}) = \frac{1}{Z_p} \exp\left[\sum_{F \in \mathcal{F}_p} \boldsymbol{\beta}_{F \to p}(y_p)\right] = \frac{1}{Z_p} \exp \sum_{F \in \mathcal{F}_p} M_F(\boldsymbol{x}_{pF}, \mathbf{d}_{pF}, y_p; \boldsymbol{\theta}_F).$$

Here $Z_p$ is the normalizer. The ideal variable marginal, for example, has the probability of 1 for the ground truth class and 0 for the remaining classes. Here we consider the cross entropy loss between the ideal marginal and the estimated marginal.

$$\begin{aligned} J(\boldsymbol{x}, \hat{\boldsymbol{y}}; \boldsymbol{\theta}) &= -\sum_{p \in \mathcal{N}} \sum_{y_p=1}^{K} \delta(y_p = \hat{y}_p) \log P(y_p|\boldsymbol{x}; \boldsymbol{\theta}) \\ &= -\sum_{p \in \mathcal{N}} \sum_{y_p=1}^{K} \delta(y_p = \hat{y}_p) \log \frac{\exp \sum_{F \in \mathcal{F}_p} M_F(\boldsymbol{x}_{pF}, \mathbf{d}_{pF}, y_p; \boldsymbol{\theta}_F)}{\sum_{y_p'} \exp \sum_{F \in \mathcal{F}_p} M_F(\boldsymbol{x}_{pF}, \mathbf{d}_{pF}, y_p'; \boldsymbol{\theta}_F)}, \end{aligned} \tag{16}$$

in which $\hat{y}_p$ is the ground truth label for the variable node $p$. Given a set of $N$ training images and label masks, the optimization problem for learning the message estimator network is:

$$\min_{\boldsymbol{\theta}} \frac{\lambda}{2} \|\boldsymbol{\theta}\|_2^2 + \sum_{i=1}^{N} J(\boldsymbol{x}^{(i)}, \hat{\boldsymbol{y}}^{(i)}; \boldsymbol{\theta}). \tag{17}$$

The work in [8] proposed to learn the variable-to-factor message ($\boldsymbol{\beta}_{p \to F}$). Unlike their approach, we aim to learn the factor-to-variable message ($\boldsymbol{\beta}_{F \to p}$), for which we are able to naturally formulate the variable marginals, which is the ultimate goal for prediction, as the training objective. Moreover, for learning $\boldsymbol{\beta}_{p \to F}$ in their approach, the message estimator will depend on all neighboring nodes (connected by any factors). Given that variable nodes will have different numbers of neighboring nodes, they only consider a fixed number of neighboring nodes (e.g., 20) and concatenate their features to generate a fixed-length feature vector for classification. In our case for learning $\boldsymbol{\beta}_{F \to p}$, the message estimator only depends on a fixed number of neighboring nodes (connected by one factor), thus we do not have this problem. Most importantly, they learn message estimators by training traditional probabilistic classifiers (e.g., simple logistic regressors) with hand-craft features, and in contrast, we train deep CNNs in an end-to-end learning style without using hand-craft features.

### 3.4 Message learning with inference-time budgets

One advantage of message learning is that we are able to explicitly incorporate the expected number of inference iterations into the learning procedure. The number of inference iterations defines the learning sequence of message estimators. This is particularly useful if we aim to learn the estimators which are capable of high-quality predictions within only a few inference iterations. In contrast,

**Table 1:** Segmentation results on the PASCAL VOC 2012 "val" set. We compare with several recent CNN based methods with available results on the "val" set. Our method performs the best.

| method | training set | # train (approx.) | IoU val set |
|---|---|---|---|
| ContextDCRF [3] | VOC extra | 10k | 70.3 |
| Zoom-out [17] | VOC extra | 10k | 63.5 |
| Deep-struct [2] | VOC extra | 10k | 64.1 |
| DeepLab-CRF [9] | VOC extra | 10k | 63.7 |
| DeepLap-MCL [9] | VOC extra | 10k | 68.7 |
| BoxSup [18] | VOC extra | 10k | 63.8 |
| BoxSup [18] | VOC extra + COCO | 133k | 68.1 |
| ours | VOC extra | 10k | 71.1 |
| ours+ | VOC extra | 10k | **73.3** |

conventional potential function learning in CRFs is not able to directly incorporate the expected number of inference iterations.

We are particularly interested in learning message estimators for use with only one message passing iteration, because of the speed of such inference. In this case it might be preferable to have large-range neighborhood connections, so that large range interaction can be captured within one inference pass.

# 4   Experiments

We evaluate the proposed CNN message learning method for semantic image segmentation. We use the publicly available PASCAL VOC 2012 dataset [19]. There are 20 object categories and one background category in the dataset. It contains $1464$ images in the training set, $1449$ images in the "val" set and $1456$ images in the test set. Following the common practice in [20, 9], the training set is augmented to $10582$ images by including the extra annotations provided in [21] for the VOC images. We use intersection-over-union (IoU) score [19] to evaluate the segmentation performance. For the learning and prediction of our method, we only use one message passing iteration.

The recent work in [3] (referred to as ContextDCRF) learns multi-scale fully convolutional CNNs (FCNNs) for unary and pairwise potential functions to capture contextual information. We follow this CRF learning method and replace the potential functions by the proposed message estimators. We consider 2 types of spatial relations for constructing the pairwise connections of variable nodes. One is the "surrounding" spatial relation, for which one node is connected to its surround nodes. The other one is the "above/below" spatial relation, for which one node is connected to the nodes that lie above. For the pairwise connections, the neighborhood size is defined by a range box. We learn one type of unary message estimator and 3 types of pairwise message estimators in total. One type of pairwise message estimator is for the "surrounding" spatial relations, and the other two are for the "above/below" spatial relations. We formulate one network for one type of message estimator.

We formulate our message estimators as multi-scale FCNNs, for which we apply a similar network configuration as in [3]. The network $C^{(1)}$ (see Sec. 3.2 for details) has 6 convolution blocks and $C^{(2)}$ has 2 fully connected layers (with $K$ output units). Our networks are initialized using the VGG-16 model [22]. We train all layers using back-propagation. Our system is built on MatConvNet [23].

We first evaluate our method on the VOC 2012 "val" set. We compare with several recent CNN based methods with available results on the "val" set. Results are shown in Table 1. Our method achieves the best performance. The comparing method ContextDCRF follows a conventional CRF learning and prediction scheme: they first learn potentials and then perform inference based on the learned potentials to output final predictions. The result shows that learning the CNN message estimators is able to achieve similar performance compared to learning CNN potential functions in CRFs. Note that since here we only use one message passing iteration for the training and prediction, the inference is particularly efficient.

To further improve the performance, we perform simple data augmentation in training. We generate extra $4$ scales ($[0.8, 0.9, 1.1, 1.2]$) of the training images and their flipped images for training. This result is denoted by "ours+" in the result table.

**Table 2:** Category results on the PASCAL VOC 2012 test set. Our method performs the best.

| method | mean | aero | bike | bird | boat | bottle | bus | car | cat | chair | cow | table | dog | horse | mbike | person | potted | sheep | sofa | train | tv |
|---|---|---|---|---|---|---|---|---|---|---|---|---|---|---|---|---|---|---|---|---|---|
| DeepLab-CRF [9] | 66.4 | 78.4 | 33.1 | 78.2 | 55.6 | 65.3 | 81.3 | 75.5 | 78.6 | 25.3 | 69.2 | 52.7 | 75.2 | 69.0 | 79.1 | 77.6 | 54.7 | 78.3 | 45.1 | 73.3 | 56.2 |
| DeepLab-MCL [9] | 71.6 | 84.4 | **54.5** | **81.5** | 63.6 | 65.9 | 85.1 | 79.1 | 83.4 | 30.7 | 74.1 | 59.8 | 79.0 | 76.1 | **83.2** | 80.8 | **59.7** | 82.2 | 50.4 | 73.1 | 63.7 |
| FCN-8s [16] | 62.2 | 76.8 | 34.2 | 68.9 | 49.4 | 60.3 | 75.3 | 74.7 | 77.6 | 21.4 | 62.5 | 46.8 | 71.8 | 63.9 | 76.5 | 73.9 | 45.2 | 72.4 | 37.4 | 70.9 | 55.1 |
| CRF-RNN [1] | 72.0 | 87.5 | 39.0 | 79.7 | **64.2** | 68.3 | 87.6 | 80.8 | **84.4** | 30.4 | 78.2 | **60.4** | 80.5 | 77.8 | 83.1 | 80.6 | 59.5 | 82.8 | 47.8 | 78.3 | 67.1 |
| ours | **73.4** | **90.1** | 38.6 | 77.8 | 61.3 | **74.3** | **89.0** | **83.4** | 83.3 | **36.2** | **80.2** | 56.4 | **81.2** | **81.4** | 83.1 | **82.9** | 59.2 | **83.4** | **54.3** | **80.6** | **70.8** |

**Table 3:** Segmentation results on the PASCAL VOC 2012 test set. Compared to methods that use the same augmented VOC dataset, our method has the best performance.

| method | training set | # train (approx.) | IoU test set |
|---|---|---|---|
| ContextDCRF [3] | VOC extra | 10k | 70.7 |
| Zoom-out [17] | VOC extra | 10k | 64.4 |
| FCN-8s [16] | VOC extra | 10k | 62.2 |
| SDS [20] | VOC extra | 10k | 51.6 |
| DeconvNet-CRF [24] | VOC extra | 10k | 72.5 |
| DeepLab-CRF [9] | VOC extra | 10k | 66.4 |
| DeepLab-MCL [9] | VOC extra | 10k | 71.6 |
| CRF-RNN [1] | VOC extra | 10k | 72.0 |
| DeepLab-CRF [25] | VOC extra + COCO | 133k | 70.4 |
| DeepLab-MCL [25] | VOC extra + COCO | 133k | 72.7 |
| BoxSup (semi) [18] | VOC extra + COCO | 133k | 71.0 |
| CRF-RNN [1] | VOC extra + COCO | 133k | 74.7 |
| ours | VOC extra | 10k | 73.4 |

We further evaluate our method on the VOC 2012 test set. We compare with recent state-of-the-art CNN methods with competitive performance. The results are described in Table 3. Since the ground truth labels are not available for the test set, we evaluate our method through the VOC evaluation server. We achieve very competitive performance on the test set: 73.4 IoU score[1], which is to date the best performance amongst methods that use the same augmented VOC training dataset [21] (marked as "VOC extra" in the table). These results validate the effectiveness of direct message learning with CNNs. We also include a comparison with methods which are trained on the much larger COCO dataset (around 133K training images). Our performance is comparable with these methods, even though we make use of many fewer training images.

The results for each category is shown in Table 2. We compare with several recent methods which transfer layers from the same VGG-16 model and use the same training data. Our method performs the best for 13 out of 20 categories.

## 5   Conclusion

We have proposed a new deep message learning framework for structured CRF prediction. Learning deep message estimators for the message passing inference reveals a new direction for learning deep structured model. Learning CNN message estimators is efficient, which does not involve expensive inference steps for gradient calculation. The network output dimension for message estimation is the same as the number of classes, which does not increase with the order of the potentials, and thus CNN message learning has less network parameters and is more scalable in the number of classes compared to conventional potential function learning. Our impressive performance for semantic segmentation demonstrates the effectiveness and usefulness of the proposed deep message learning. Our framework is general and can be readily applied to other structured prediction applications.

**Acknowledgements** This research was supported by the Data to Decisions Cooperative Research Centre and by the Australian Research Council through the ARC Centre for Robotic Vision CE140100016 and through a Laureate Fellowship FL130100102 to I. Reid. Correspondence should be addressed to C. Shen.

## Footnotes

[1] The result link provided by VOC evaluation server: http://host.robots.ox.ac.uk:8080/anonymous/DBD0SI.html

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
