[Supplementary Material · supplementary.pdf]

# Supplementary Document:
# Deeply Learning the Messages in Message Passing Inference

**Guosheng Lin, Chunhua Shen, Ian Reid, Anton van den Hengel**
The University of Adelaide, Australia; and Australian Centre for Robotic Vision
E-mail: {guosheng.lin,chunhua.shen,ian.reid,anton.vandenhengel}@adelaide.edu.au

This document contains the following contents:

1. Figure 1 shows the illustration of Network $C^{(1)}$ for generating multi-scale feature map.
2. Figure 2 shows the illustration of Network $C^{(2)}$ for generating message vector.
3. Figure 3 shows the illustration of building pairwise connections.
4. Figure 4 shows the network layer configuration.
5. Figure 5 shows some prediction examples of our method for semantic segmentation.

Figure 1: Illustration of Network $C^{(1)}$ for generating multi-scale feature map.

Figure 2: Illustration of Network $C^{(2)}$ for generating message vector.

Two types of spatial relations
for pairwise connections

Final feature map

Build factor graph

1. Create one node for each spatial
position in the feature map

2. A node is connected to all other
nodes which lie within the dash box

Surrounding relation

Above/Below relation

Figure 3: Illustration of building pairwise connections. We denote by $a$ the length of the shortest edge of the feature map. The size of the range box (dash box in the figure) size is $0.4a \times 0.4a$. For the surrounding relation, the range box is centered on the node. For the above/below relation, the bottom edge of the range box is centered on the node.

Network C(1)
(6 convolution blocks)

| Block 1: | Block 2: | Block 3: | Block 4: | Block 5: | Block 6: |
|---|---|---|---|---|---|
| 3 x 3 conv 64 | 3 x 3 conv 128 | 3 x 3 conv 256 | 3 x 3 conv 512 | 3 x 3 conv 512 | 7 x 7 conv 4096 |
| 3 x 3 conv 64 | 3 x 3 conv 128 | 3 x 3 conv 256 | 3 x 3 conv 512 | 3 x 3 conv 512 | 3 x 3 conv 512 |
| 2 x 2 pooling | 2 x 2 pooling | 3 x 3 conv 256 | 3 x 3 conv 512 | 3 x 3 conv 512 | 3 x 3 conv 512 |
|  |  | 2 x 2 pooling | 2 x 2 pooling | 2 x 2 pooling |  |

Network C(2)
(2 fully-connected layers)

Fully-conn 512
Fully-conn K (number of classes)

Figure 4: Network layer configuration. The top 5 convolution blocks share the same configuration as the convolution blocks in the VGG-16 network. The stride of the last max pooling layer is 1, and for other max pooling layers we use the same stride setting as that of the VGG-16 network.

(a) Testing     (b) Truth     (c) Prediction

(d) Testing     (e) Truth     (f) Prediction

Figure 5: Some prediction examples of our method.