[Reviews · NeurIPS 2015]

Submitted by Assigned_Reviewer_1

Line 91: joint -> jointly Line 161: massage -> message Line 266: convlutional -> convolutional Line 269: correspond -> corresponds Line 301: reminding -> remaining Line 408: We has -> We have Line 416: perform -> performs
Summary: The authors present a solid idea and demonstrate that it obtains a modest improvement over state-of-the-art on PASCAL 2012 segmentation. The idea is interesting however it is not particularly well presented in the text. I would recommend the use of figures in the exposition.

Submitted by Assigned_Reviewer_2

This paper presents a CNN-CRF approach for structured prediction by explicitly learning CNN messages for inference. I have several concerns as described below:

(1) A schematic figure might help explain the overall method. So far the exact roles of the proposed components are not clear. The equations should be made more intuitive.

(2) The scope of the proposed method seems to be quite limited as only very limited number of neighborhood connects can be considered and there are many approximations to the method. It lacks sufficient elegance and cleanness in the algorithm.

(3) The experimental results are only reported on the VOC dataset. Since this is a NIPS submission, justification on other popular datasets for structured labeling, e.g. OCR and POS, is needed.

Overall, the method is quite convoluted and it is hard to see what exactly the benefit of the proposed method is.
Summary: The general direction of tackling the structured prediction problem by learning message passing in convolutional neural networks based CRF model is good. Improved results over the state-of-the-art on the PASCAL VOC image labeling dataset are reported. However, the scope of the proposed method is till somewhat limited and the solution is kind of intermediate.

Submitted by Assigned_Reviewer_3

This paper proposes to directly learn the messages in message-passing inference (as in inference machines[8]) using CNNs. The paper shows state-of-the-art results on PASCAL VOC semantic segmentation.

First, note that [12] also learn messages in one-step message-passing inference. The math is not described well, and the details are different (that paper tackles pose estimation) but the insight that learning messages is easier than learning potentials, and that CNNs can be used to learn messages, is similar. This paper is missing an acknowledgement of this, and a discussion of what this paper adds.

Second, let's take a closer look at what the approach is actually doing: the final score vector for a pixel is the sum of the scores output by each factor->pixel message, and each factor->pixel message is computed by a *separate deep network*. This means that the method is eerily close to a simple ensemble of four models trained with different initializations.

All the other methods compared to (except ContextDCRF, which suffers from a similar problem) have only one network making the prediction. This suggests that an ensemble model is a natural baseline. I would encourage the authors to try such a model.

Given these two points I am somewhat skeptical of the originality and significance. In terms of clarity, the paper is well written, and the math is very clearly described. I would still recommend the paper for acceptance because this idea of learning messages using CNNs is important, and because the results are state-of-the-art.

Summary: The notion that one can train a CNN to predict the messages is quite interesting and is described well. However, [12] also directly learn the messages (for a one-step message passign inference) using CNNs, albeit for a different task. The ensemble effect coming from multiple networks is also not disentangled (see below)

Submitted by Assigned_Reviewer_4

Given CRF potentials, one can write down the marginal predictions after only passing the initial messages from factors to variables. This paper incorporates that parametric form into a neural network, and fits it directly to minimize training error. This architecture appears to be empirically successful on a meaningful benchmark. I found the framing of the work partially misleading: As far as I can tell, the cost function reported doesn't care about structured prediction just pixel-wise errors, and no CRF model is actually fitted.

To me "structured CRF prediction" means that there is a joint distribution over the labels given an input. Given such a model, it would be possible to fantasize joint plausible configurations of the labels, or report the jointly most probable labeling.

I don't think the "Intersection over Union" score cares if the vector of labels is jointly plausible. The way to maximize this score is to predict the most probable label for each pixel independently. Learning a CRF model would learn a joint distribution, but here predictive performance, which doesn't care about the joint distribution, is maximized, not a CRF model. It isn't even clear to me if the "messages" in this paper will always correspond to realizable factors(?), let alone likely ones.

The work is certainly interesting. The idea is a neat one, and seems to be an excellent heuristic for setting the parametric form of the output layer of a neural network to get good performance. I simply found some of the framing misleading and genuinely confusing on first reading.

line 332: "One advantage of message learning is that we are able to explicitly incorporate the expected number of inference iteration into the learning procedure." -- However, only one iteration is ever tried, and I presume implemented. I'm not convinced the details have been worked out or investigated enough to make this claim.

Typo: line 54: massage -> message

There's quite a few incorrect word endings, usually singular/plural disagreement. I lost motivation to note them down after noticing a spell checker hadn't been run: line 66: potenials line 120: sematic line 265: convlutional
Summary: A neat way to set the output layer of a neural network for labelling pixels. Inspired by the parametric form of CRF messages, but not actually fitting a structured prediction model (as advertised) as far as I can tell.

Nothing in the rebuttal changes my view: this paper isn't doing structured prediction as it claims it does. It does appear to be accurate at pixel-wise labelling.

Author Feedback
Author rebuttal: All reviewers agree that the proposed idea is interesting.
We set new state-of-the-art results on semantic image segmentation.

Thanks for the reviewers' efforts to make the draft better.

R1:

Q1: A schematic figure?

A: Thanks for the suggestion, we will add illustrative figures in the final version.

Q2: very limited number of neighborhood connects?

A: This is not true. We do not have this limitation.
We follow the graph connection setting in [3].
One node typically connects to around 100 nodes that lie in a spatial region.

Q3: More structured labeling experiments?

A: Currently we focus on semantic image segmentation on the challenging PASCAL VOC dataset.
The proposed deep message learning approach is general, which can be readily applied to other applications.

R2:

Q1: [12] also learn messages?

A: We will certainly add more discussion about the similarities/differences with [12] in the final version. Though the math in [12] is not very clear (as noted by the reviewer), the concept of learning message estimators is not directly discussed in [12]. The approach [12] proposes a "Spatial-Model" for pose estimation.
They learn the marginal likelihood of body parts but ignore the partition function in the likelihood.

Q2: suggestion on comparing to ensemble of four models trained with different initialization?

A: We will verify this according to the reviewer's suggestion.
We suspect that training 4 CNN unary models with different initialization would lead to minor improvements, since these models might not able to show significant diversity. In contrast our message estimators are able to capture contextual information from different types of spatial relations.

R3:

Q1: no CRF model is actually fitted?

A: We do not directly learn the potentials in CRF model,
which is different from conventional CRF learning approaches.
We target inference-based structured prediction, using message passing, relying on the message passing to output the final structured prediction.
Conventional CRF approaches calculate the messages from potentials (factors), but -- in contrast -- we learn CNNs to directly output the required messages.

R4:

Q1: original contribution? compared to related papers [3] and [8]?

A: We have discussed [3] and [8] in the related work and experiment sections.
Basically, the recent work [3] follows the conventional CRF learning scheme which learns CNN based potential functions. In contrast, we directly learn CNN based message estimators instead of potential functions, which can be seen as the important extension of the CVPR2011 paper [8] for efficient learning of deep structured models.

Q2: minor difference between learning potentials functions in Eq(5) and learning message estimator?

A: As discussed in Sec.2 (page 3),
directly learning the potentials in Eq(5) brings the challenges for training.
CNN training usually requires a huge number of SGD iterations (hundreds of thousands, or even
millions). Directly learning the potentials requires marginal inference in each SGD iteration,
which make the training become intractable, even just doing a few iterations of message passing.
In our approach for learning message estimator, we don't need to call marginal inference at each SGD iteration, which is much more efficient and practical for large-scale learning.
Moreover, the message estimator network for has only K (the number of classes) output dimensions (see Sec.1 on page 2), independent to the order of factors. We will include the discussion of Bethe-Free energy.

Q3: graph structure details?

A: Thanks for the suggestion, we will include more illustrative figures in the final version.

Q4: clearly heuristic? theoretical statements about convergence or exactness?

A: We think a characterisation of our method as "clearly heuristic" is a bit unfair. However we acknowledge that we do not have theoretical guarantees of convergence; we will certainly make this clearer in the introduction as requested by the reviewer.

R5 and R6:
Thanks for the suggestions and typo corrections.
We will add more illustrative figures in the final version